# Inhibitory Plasticity: From Molecules to Computation and Beyond

**DOI:** 10.3390/ijms21051805

**Published:** 2020-03-06

**Authors:** Daniela Gandolfi, Albertino Bigiani, Carlo Adolfo Porro, Jonathan Mapelli

**Affiliations:** 1Department of Biomedical, Metabolic and Neural Sciences and Center for Neuroscience and Neurotechnology, University of Modena and Reggio Emilia, Via Campi 287, 41125 Modena, Italy; daniela.gandolfi82@gmail.com (D.G.); albertino.bigiani@unimore.it (A.B.); carlo.porro@unimore.it (C.A.P.); 2Department of Brain and behavioral sciences, University of Pavia, 27100 Pavia, Italy

**Keywords:** synaptic plasticity, inhibition, computational neuroscience, GABA, LTP, LTD

## Abstract

Synaptic plasticity is the cellular and molecular counterpart of learning and memory and, since its first discovery, the analysis of the mechanisms underlying long-term changes of synaptic strength has been almost exclusively focused on excitatory connections. Conversely, inhibition was considered as a fixed controller of circuit excitability. Only recently, inhibitory networks were shown to be finely regulated by a wide number of mechanisms residing in their synaptic connections. Here, we review recent findings on the forms of inhibitory plasticity (IP) that have been discovered and characterized in different brain areas. In particular, we focus our attention on the molecular pathways involved in the induction and expression mechanisms leading to changes in synaptic efficacy, and we discuss, from the computational perspective, how IP can contribute to the emergence of functional properties of brain circuits.

## 1. Introduction

The emergence of brain functions, from motion control to cognition and abstract thinking, is tightly bound to the ability of brain circuits to adjust synaptic connections [1]. For a long time, this capability was hypothesized to rely exclusively on the adaptability of excitatory synapses, assuming the substantial invariance of inhibitory connections. Only recently, a wide number of molecular and cellular mechanisms residing at inhibitory synapses and responsible for the emergence of complex brain states are beginning to be unraveled [2]. It is, in fact, evident that inhibitory synapses throughout the brain exhibit activity-dependent changes of their connectivity weights both in the form of long-term potentiation (LTP) and long-term depression (LTD). Nevertheless, the investigation of activity-dependent changes of inhibitory synapses has been traditionally prevented, or strongly limited by the wide number of GABAergic cell types and the consequent difficulty of isolating specific neuronal pathways [3]. The extraordinary variety of inhibitory interneurons and related connections expands the number of plasticity subtypes, which can be expressed. Importantly, the impact of IP on neuronal circuits, by acting on the overall neuronal excitability and on the possibility to further induce persistent forms of plasticity, can have important consequences on the brain functionality [4]. There is indeed increasing attention on the various forms of IP since circuit refinement induced by changes of the Excitatory/Inhibitory (E/I) balance can strongly influence learning and memory. In particular, the advent of sophisticated techniques either in the form of molecular, genetic, or electrophysiological and imaging approaches has started to allow precise dissection of microcircuits and of single synaptic types [5]. Moreover, long-term changes in inhibitory activity can induce pathological alterations of brain functions, while several neuropsychiatric disorders have been shown to be related to permanent GABAergic dysregulations [6,7].

In this article, we discuss how the various forms of IP shape brain circuits architectures and functions with particular attention on neuronal computation. We begin with a panoramic view on the variety of GABAergic cell types and circuits to continue with an analysis of the induction and expression mechanisms of inhibitory LTP and LTD. We then proceed with discussing the functional consequences of the aforementioned mechanisms, including computational models proposed to predict the effects of IP. Finally, we analyze the potential involvement of inhibitory plasticity in the emergence of pathological disorders to end with a few suggestions regarding the implementation of synaptic learning rules into artificial circuits. 

## 2. Variety of Inhibitory Circuits in the Central Nervous System

Although it is widely accepted that inhibitory neurons are actively engaged in network computation by providing global stability to network dynamics, by controlling the degree of circuit synchronization and by controlling the timing of neuronal firing [8], it is still under debate the way fine-tuning of inhibitory connections participate into regulatory mechanisms of circuit dynamics. Certainly, information processing is strictly dependent on how excitation and inhibition are in balance with each other, engaging directional and recurrent wired networks, implementing computational functions like the expansion of the dynamic range of neuronal responses [8], the input separation through winner-take-all schemes [9], or spatial pattern separation through combinatorial operators [10]. Nevertheless, even the lack of a precise definition of the concept of balance strongly limits the possibility to fully understand the interplay between excitatory and inhibitory signaling. For instance, the temporal and spatial scales over which neuronal activity is tuned by the interplay between glutamatergic and GABAergic synapses affect the timing of spike generation [11,12,13] rather than the average firing rate [14] or the synchronization of local [15] and global networks [16].

One of the major obstacles in identifying the way inhibitory plasticity tunes the activity of brain circuits comes from the diversity of inhibitory interneurons in the CNS, which still leaves these cells the capacity to provide inhibition to a large variety of excitatory input classes [3]. Among neuronal categories, almost 20% is GABAergic [17] and show a wide variety of functional and molecular subtypes with variable locations and, interestingly, capacities for plasticity [2,18,19,20]. In some cases, even the classification of interneurons as exclusively inhibitory is technically difficult because, in early development, GABA can act as depolarizer [21], and into adulthood, some axo-axonic contacts may continue with this behavior [22].

Several attempts have been indeed made to classify cortical interneurons, for instance, the so-called “Petilla terminology” [23], by collecting features describing interneurons, classifies GABAergic cells following (i) morphological and (ii) molecular properties. A major effort has been made indeed to generate clusters of inhibitory neurons based on their gene expression. Although an attempt in classifying interneurons has been made by considering single properties independently, the amount of subtypes emerging from considering all the possible combinations increases dramatically. More in detail, recent works have identified at least 10 distinct classes of inhibitory neurons in the hippocampal circuits [24] with more than 30 subclasses. Although most of them show overlapping functional and computational properties, the analysis of the contribution of inhibitory neurons and plasticity in circuit computation has to deal with this heterogeneity. When focusing the attention only on cortical networks, the variety of GABAergic neurons has been investigated under the molecular, morphological, and functional points of view, as well as on the ability to undergo synaptic plasticity [3]. Furthermore, a critical issue regards the computational capacity of interneurons deriving from the organization of synaptic connectivity. For instance, chandelier cells, interneuron with the anatomical property of embracing the hillock of target neurons, can implement a simple modulation of the action potential generation in principal neurons (PNs) by exploiting synaptic contacts on the initial tract of axonal segments [25]. Conversely, basket cells provide inhibition to cell bodies and proximal dendrites of PNs. The strategic location of efferents and the confined segregation of synapses allow controlling spike timing, oscillations, and integrative functions such as orientation selectivity or refinement of sensory maps [26] by exploiting peri-somatic innervation. Back to the molecular distinction, the neuronal subtypes identified in GABAergic interneuron classes can be, however, resumed into four main groups resulting from the non-overlapping neuronal clustering correlated to the expression of different markers: a) ParValbumin (PV), b) cholecystokinin (CCK) c) the co-transmitter SOMatostatin (SOM) and d) serotonin receptor 5-HT3A (5-HT). The main electrophysiological feature of PV cells is a fast-spiking firing enabling a powerful control of timing and rate of spike output from postsynaptic neurons [27]. Furthermore, PVs interneurons are the main actors in driving oscillations in cortical circuits [28,29]. The CCK interneurons, particular inhibitory subtypes with characteristics similar to PVs cells in terms of the anatomical organization of efferents, show regular firing providing fine control of the postsynaptic activity of PNs of both neocortical and hippocampal regions [30]. Even though CCK and PV neurons show differences, they share similar peri-somatic inhibitory properties, hence allowing the clustering in the same functional group. This aspect is particularly important in terms of the computational effects of IP because the morphological differences between neurons belonging to the same molecular class are responsible for the heterogeneity in the firing patterns. The peculiar electrophysiological characteristics such as membrane time constant, membrane capacity, and resistance, as well as leakage or conductance are critically involved in synaptic integration. Differently from PVs and CCKs, SOM neurons preferentially contact dendrites either on spines or shafts [31]. The main characteristic of SOM inhibitory interneurons are their action against the spatio-temporal diffusion of signals in dendrites [32]. By exerting their activity at dendritic level, SOM neurons regulate i) dendritic calcium fluxes, which are in turn involved into the induction mechanisms of LTP and LTD [33], ii) the insurgence of dendritic spikes driving neurons to somatic bursting activity and iii) other forms of stereotyped network activity [34]. The last class of inhibitory neurons (5-HT3A) can be subdivided into neuroglia form, inhibiting dendrites of excitatory neurons and thus massively suppressing circuit activity [35] and cells expressing vasoactive intestinal peptide (VIP) which exert their actions mainly versus other inhibitory circuits with paradoxical excitatory consequences on the overall network activity [36]. It appears evident that, despite the wide spectrum of molecular and morphological subtypes, inhibition is mainly exploited by the anatomical organization of afferent contacts, which can be alternatively perisomatic or dendritic. This distinction, in turn, leads to the determination of functional properties that can result in the emerge of the mechanisms leading to long-term plasticity induction and expression.

When considering the functional differences between GABAergic classes, it is evident that the shown diversity can be one of the main causes of the inhibitory capability to stabilize and finely regulate circuits activity [37]. Furthermore, since inhibitory neurons create a wide network of electrically and synaptically coupled cells with an exceptional variety of physiological and anatomical properties, it is widely accepted that inhibition cannot be merely considered as a regulator of circuit excitability [38,39]. Additionally, the reciprocal and recurrent broad connections between inhibitory and excitatory neurons are ideal to condition large circuits areas [40]. Several GABAergic neurons, in fact, widely connect excitatory cells while locally contact inhibitory neurons exploiting an interplay between excitation and inhibition in broad neural circuits [41], which is essential to maintain the circuit balance that favors neural computation [42].

## 3. Induction and Expression Mechanisms

Given the diversity of inhibitory classes, it can be envisaged that inhibitory plasticity presents heterogeneous molecular and functional characteristics accordingly. Interestingly, as observed for excitatory synapses, IP was shown to occur as changes in the presynaptic release, in postsynaptic GABA_A_ receptors (GABA_A_Rs) activity or in mixed forms [43]. Modifies at the presynaptic side require retrograde signaling that can persistently modulate GABA release [44], whereas purely postsynaptic mechanisms involve alterations of GABA receptors machinery [45]. The modulation of neurotransmitter vesicles release from presynaptic boutons is mainly triggered by heterosynaptic mechanisms, thus requiring non-GABAergic stimuli [44,46] from nearby synapses, and actually, the activation of inhibitory fibers is indeed not required. In order for this to happen, signals must be communicated to the presynaptic terminals following the postsynaptic induction, which can involve a wide series of mechanisms. The easiest and the most used strategy is via diffusible molecules acting as retrograde messengers [47]. Alternatively, the glutamate released during the induction process can be directly spread toward GABAergic terminals to induce changes in vesicles release through the activation of presynaptic receptors [48]. In the majority of the reported cases, heterosynaptic inhibitory plasticity was induced by high-frequency or theta-burst stimulation of excitatory axonal terminals [49]. The first forms of potentiation presynaptically expressed through the glutamatergic action of excitatory fibers were reported in the primary visual cortex and in the cerebellum. The high-frequency stimulation of layer 4 induced LTP of GABAergic inhibitory postsynaptic potentials (IPSPs) in layer 5 pyramidal neurons [50], through a not well-identified mechanism involving N-methyl-D-aspartate (NMDA) receptors in both the pre- and postsynaptic components. Interestingly, this form of IP remains one of the most investigated forms of plasticity given its importance in the determination of the E/I balance in the developing visual cortex [51]. Similarly, the stimulation of cerebellar glutamatergic climbing fibers, bringing the teaching error for motor learning to occur, induces a calcium-dependent long-lasting potentiation of inhibitory postsynaptic currents (IPSCs) in Purkinje cells mediated by molecular layer interneuron [52]. In the subsequent years, heterosynaptic LTP at the inhibitory connections has been discovered in several other brain areas, including the Ventral Tegmental Area [44], lamina I of the spinal cord [53], neonatal hippocampus [54] and basolateral amygdala [55]. Interestingly, heterosynaptic inhibitory potentiation shows strong similarities with the classical form of LTP discovered in the hippocampus in the early 70s: synapse specificity, associativity, calcium signaling, and dependence on NMDA receptors activation [56]. As in the case of presynaptic excitatory LTP [57], the potentiation of inhibitory synapses reported so far requires signals to be conveyed to presynaptic terminals following postsynaptic induction (Figure 1). The retrograde messengers, a class of molecules produced in the postsynaptic cell in an activity-dependent manner and traveling backward, are essential to modulate neurotransmitter release, therefore, allowing the expression of LTP. Various pathways involving retrograde messengers and participating in the triggering of inhibitory plasticity have been identified; however, two main molecules modulating heterosynaptic LTP are more recurrently found in different brain regions: i) the diffusible nitric oxide (NO) [44,46] and the brain-derived neurotrophic factor (BDNF) [58,59]. The NO originates in the postsynaptic compartment following calcium entry (Figure 1B). The activation of the Nitric Oxide Synthase (NOS) following postsynaptic calcium rise catalyzes NO, which can freely diffuse in the extracellular matrix by virtue of its gaseous nature. The NO then triggers cGMP, and eventually, other molecular targets in the presynaptic boutons [60]. The BDNF is involved in the regulation of neurogenesis, activity-dependent synaptic plasticity, and in other non-neuronal mechanisms [61]. It binds to its high affinity tyrosine kinase B (TrkB) receptor to activate transduction cascades crucial for early gene expression [62]. The BDNF signal cascade can be triggered by several mechanisms (Figure 1C), including calcium influx through voltage-dependent channels [63] and GABA_B_ receptors activation [64]. In an alternative way to the retrograde diffusion of postsynaptically synthesized molecules, IP could be triggered, as in cerebellar stellate cells, by the direct activation of NMDA receptors (Figure 1D) on presynaptic GABAergic terminals in response to the glutamate released from parallel fibers [65]. Similarly, in the frontal cortex of developing rats, the calcium influx through NMDA-Rs opening caused by glutamate diffusion from nearby synapses is sufficient to trigger the increase of GABA release [48]. It should also be noted that a particular mechanism observed for inhibitory LTP has been characterized in the dorsomedial hypothalamus. The activation of CKK receptors by the exposure to neuromodulators combined with the concomitant activation of metabotropic glutamate receptors (mGluRs) induces the release of Adenosine triphosphate (ATP) by surrounding astrocytes acting on presynaptic receptors and in turn triggering a prolonged increase in GABA release [66]. 

As in the case of i-LTP, the identified forms of i-LTD presynaptically expressed depend either on the direct NMDA-Rs activation in the GABAergic terminals [67] (Figure. 2) or through the retrograde diffusion of endocannabinoid (eCB) [49] (Figure 2D). These molecules, in response to afferent fibers stimulation, move from the post- to the presynaptic terminal triggering LTD induction. The eCB-dependent i-LTD, which is widely expressed throughout the brain [68,69,70,71], often requires the spread of glutamate from nearby excitatory synapses to activate metabotropic receptors (mGluR) as a trigger (Figure 2D). Interestingly, this form of plasticity does not necessarily involve calcium influx in the postsynaptic terminals, as demonstrated for the mGluR-dependent inhibitory LTD (i-LTD) in the amygdala [72]. Conversely, hippocampal i-LTD mediated by eCB is triggered by interneuron activity likely bringing calcium increase through voltage-dependent calcium channels, which in turn induces the simultaneous enhancement of calcineurin activity and the consequent reduction of the adenylyl cyclase-protein kinase A (PKA) transduction cascade leading to a long-term decrease of GABA release [73]. Additionally, other factors may actually contribute to modulate the retrograde diffusion of eCB. For instance, the dopamine receptors type 2 (D2R) were shown to suppress GABA release in the prefrontal cortex [74] and in the Ventral Tegmental Area [75] through a coactivation of the D2R and Cannabinoid Receptors by an increase of endogenous dopamine levels. As in the case of inhibitory presynaptic LTP, also the depression of GABA release can be induced by the direct activation of presynaptic NMDA-Rs by glutamate released from excitatory synapses in the next proximity [46,76]. Interestingly, our recent results show that the cerebellar inhibitory synapse between Golgi and Granule cells simultaneously exploits these mechanisms. The theta-burst protocol delivered to the excitatory mossy fibers can bidirectionally modulate GABA release through glutamate diffusion inducing LTP through the retrograde diffusion of nitric oxide toward GABAergic synapses or, alternative, LTD can be triggered by presynaptic activation of NMDA receptors [46].

Inhibitory plasticity can also be induced through mechanisms requiring the direct activity of GABAergic afferents. One of these homosynaptic mechanisms has been described in the primary visual cortex, where the firing of a presynaptic neuron paired with the activation of a postsynaptic pyramidal neuron can induce inhibitory plasticity [77,78]. The mechanisms subtending homosynaptic plasticity can depend on calcium changes, as in the case of star GABAergic connections between fast-spiking and star pyramidal neurons in the visual cortex during visual deprivation [77]. Nevertheless, other areas such as neocortex show inhibitory plasticity strongly correlated to calcium influx elicited by paired action potential during induction protocols [78] (Figure 2C). Additionally, it has been shown that a shift in the chloride transporter altering the driving force for GABAergic currents can occur in hippocampal neurons in response to coincidence activation of pre and postsynaptic activity [79]. 

Changes in inhibitory strength can also be associated with purely postsynaptic expression through a large variety of mechanisms (Figure. 3) [80], as it happens for excitatory synapses [81]. GABAergic weights can be adjusted postsynaptically by bidirectional changes in channels functionality. The ionotropic GABA_A_ receptors, in response to specific patterns of induction requiring calcium influx following postsynaptic firing activity, can be phosphorylated by different kinases (e.g., PKC, CaMKII, Src, and PKA) [82] (Figure 3B). The high-frequency firing in neocortical pyramidal neurons drives LTP of perisomatic inhibition via calcium entry through R-type voltage-gated calcium channels [83], which can be reverted to depression requiring calcium via L-type channels during hyperpolarization [84]. A similar form of postsynaptic plasticity was reported to occur in cerebellar Purkinje cells, where the potentiation of GABA release is triggered by repetitive postsynaptic discharge [52]. Interestingly, the increase of perisomatic inhibition by cerebellar basket cells through an increase of receptors trafficking is reported to be also triggered by the sole excitatory activation of climbing fibers [52]. Another form of postsynaptic plasticity was shown to occur in the hippocampal circuits and is expressed as an increase in the expression level of the scaffold protein for GABA_A_ receptors gephyrin [85]. The availability of this molecule at GABAergic synapses is regulated by its state of phosphorylation [86] and is responsible for the induction of postsynaptic LTP that alters GABA_A_-Rs dynamics. Moreover, the continuous cycling of GABA_A_-Rs insertion and removal, together with movements of lateral diffusion at synaptic surface regulates synaptic functionality [87]. In addition, the number of GABA_A_-Rs may change in response to receptor trafficking regulation. Inhibitory responses can be, therefore, bidirectionally modulated by alternatively acting on the exocytosis and endocytosis cycling [88]. Finally, also in the case of postsynaptic mechanisms, postsynaptic changes of intracellular concentrations of membrane-permeable ions can contribute to GABergic signaling [89]. 

The activity of extrasynaptic receptors mediates an alternative form of synaptic inhibition. The impact of this form of tonic inhibition is critically related to its impact on membrane conductance and membrane potential with time constants considerably lower than the one of receptors located in the terminals [90]. The tonic inhibition has been shown to undergo several forms of plasticity too; however, in most cases, glutamatergic signaling is essential to trigger persistent changes. In the hippocampus, kainate receptors activation triggers LTP of tonic inhibition [91], whereas persistent potentiation can be induced by block or genetic deletion of NMDA receptors [92] whilst depression is triggered by the activation of NMDA receptors [93]. Furthermore, tonic inhibition can be regulated by the direct activation of CB1 receptors [94] by retrograde diffusion of NO [95] or, alternatively by the direct activation of muscarinic acetylcholine receptors [96]. The fast synaptic inhibition related to the activity of GABA_A_-Rs has been well characterized and described. Conversely, the functional alternative to the action of ionotropic GABA_A_-Rs is the slow inhibition mediated by the metabotropic GABA_B_ receptors. Although the biochemical signaling engaged by GABA_B_-Rs is characterized in detail as well as their role in shaping neuronal activity, it is not much understood whether they can undergo plastic changes. Nevertheless, reports of persistent changes in GABA_B_-Rs activity have been shown to occur in the hippocampus [97] and in lateral habenula [98], albeit the cellular and molecular mechanisms underlying this form of long-term plasticity need further investigation. It should also be noted that, as in the case of glutamatergic synapses, where the activity of G protein-activated inwardly rectifying K^+^ (GIRK) channels has been shown to induce LTP in cultured hippocampal neurons [99], in a recent work, Sanchez-Rodriguez and colleagues showed that GIRK channels are implicated in the expression of inhibitory LTP in the hippocampal circuit in vivo and, importantly, these mechanisms are impaired by the presence of amyloid-β (Aβ), raising attention on the implication of inhibitory plasticity in neurodegenerative diseases [100]. 

## 4. Learning Rules and Computational Consequences of Inhibitory Plasticity

Unlike excitatory synapses, learning rules for inhibitory plasticity have not yet been extensively investigated [101]. However, among the variety of rules, the spike-timing-dependent plasticity (STDP) correlating the reciprocal timing of pre and postsynaptic firing [102] has been encoded for some GABAergic synapses [4,103]. The first reported evidence dates back to 2001 when Holmgren and Zilberter showed that in the neocortex, when action potentials in the presynaptic inhibiting interneuron are timely proximal to postsynaptic pyramidal cell firing, GABAergic synapses undergo LTD [78]. Conversely, if presynaptic spikes and postsynaptic firing are distant enough, synaptic weights are potentiated in a calcium-dependent way. After a few works demonstrating in hippocampal circuits the presence of a few variants of inhibitory STDP [104,105,106], in 2006, Haas and colleagues showed that in stellate cells of rat entorhinal cortex presynaptic incoming before postsynaptic spikes trigger strengthening of inhibitory synapses while the reserve leads to depression [107]. Surprisingly, the maximal efficacy changes did not perfectly match the pre-post coincidence, instantiating a time-shift with important consequences in the circuit computational rules. The authors, in fact, by using a mathematical model, showed that STDP exploits the clustering of neurons within the circuit providing flexible and dynamic organization of neuronal circuitry in a region where the uncontrolled spread of excitation often leads to epileptic foci. Similarly, in pyramidal neurons of the mouse auditory cortex, STDP is exploited by inhibitory conductance. The maximal effect was observed when pre and postsynaptic spikes were temporally proximal, with a relatively large time window (≈ 10 ms), and independently from their reciprocal order [108]. Given the tendency to show potentiation with paired activity and the requirements of NMDA-Rs activation for the induction processes, this form of plasticity seems to be finalized to silencing network activity in response to diffuse circuit activation [108]. Furthermore, the auditory cortex also shows STDP in the PNs mediated by GABA_B_-Rs with LTD induced by presynaptic before postsynaptic spikes. The fact that the sign of plasticity can be reversed during development makes this form of plasticity a suitable candidate for disinhibition during the auditory critical period [109]. Differently, electrophysiological recordings in the somatosensory cortex revealed that the coupling between prolonged postsynaptic bursts with single presynaptic spikes in temporal proximity leads to GABAergic depression, whereas potentiation was observed when presynaptic spikes were presented well beyond the end of the burst [73]. This mechanism has been suggested to participate in the sharpening of significant sensory patterns. In hippocampal neurons, the sole presynaptic spikes lead to depression of synaptic conductance while the coupling of pre and postsynaptic activity generated changes in local chloride reversal potential with the same sign [79]. At the circuit level, this effect appears to weaken the inhibitory strength leading the system toward a critical large excitatory/inhibitory balance with substantial reverberations on network activity [110]. Nevertheless, the firing activity of other neurons in the circuit [111], as well as membrane potential value during induction [48], could play a significant role in shaping synaptic changes by affecting the amplitude and direction of plasticity. 

The precise identification of the functional role of inhibitory plasticity is still an open issue. Nonetheless, the recurrent leitmotif regarding GABAergic plasticity is the maintenance of a constant E/I balance in a circuit that can compensate for changes in the excitatory driving force triggered by plasticity at glutamatergic synapses. This homeostatic regulation can be obtained by reducing both feed-forward inhibition and excitation [112]. Alternatively, increasing the excitability of inhibitory interneurons following the potentiation of glutamatergic synapses can balance the circuit functioning, as it was shown in hippocampal circuits [113]. In the somatosensory cortex, the recruitment of inhibitory cells by the activity of pyramidal neurons can contribute to finely regulate cortical excitability through the sensitivity and the dynamic range of recurrent inhibition [114]. Furthermore, the tight correlation between fast inhibition and excitation allows the fine regulation and the balancing of neuronal circuits, either in the form of spontaneous or evoked firing activity [115]. Nevertheless, although the maintenance of the E/I balance seems essential for the correct circuit activity, changes in the excitatory to inhibitory balance could play a key role in receptive field organization [116] and sensory learning [117]. Importantly, since the exact value of E/I can be adjusted on different setpoints according to the brain region, the circuit activation by input stimuli can alternatively lead to the suppression or to the potentiation of excitatory output. It has recently been shown, in fact, that in neocortical circuits, the persistent increase of GABAergic synapses can impact output firing through a decreased spike probability and increased timing [118]. Furthermore, we have recently shown in cerebellar cortex that bidirectional plasticity of inhibitory circuits [46] contributes to control the spatial and temporal pattern activity of excitatory granule cells by sharpening center-surround structures [119] and by finely regulating the timing of first spike output through subthreshold integration processes [120].

From a purely computational perspective, the power of the brain is traditionally linked to the complex connectivity of neuronal networks, whereas single neurons are considered as linear integrators and thresholding devices. It is indeed clear that a wide series of non-linear mechanisms converting synaptic input into output firing is employed by single neurons to process information. These mechanisms include synaptic noise, inhibitory conductance, and notably synaptic plasticity [121]. As a general rule, the analysis of the effects of plasticity on neuronal computation has been mainly focused on excitatory circuits; nonetheless, recent discoveries on the involvement of IP require to deepen the investigation of its effects on network computation. Regardless of molecular subtypes, the prominent characteristic of inhibitory plasticity for its consequences on network computation is the architectural organization of inhibitory afferents. In particular, single neuron computation is strongly modulated by peri-somatic inhibition, which in turn exerts a critical additive or subtractive effect on the input-output ratio (I/O) when potentiated or depressed [121]. The specific peri-somatic targeting onto excitatory cortical neurons strongly influences the insurgence of network oscillations, which can be traced back to cognitive and sensory functions [122]. It has been suggested that the bidirectional modulation of peri-somatic inhibition by LTP and LTD could alternatively entrain single neurons in synchronous activity or decoupling oscillatory events [123], thus favoring the coordination of neurons sharing common functional properties. Conversely, despite dendritic inhibition is a crucial determinant for synaptic integration and underlies lateral inhibition during sensory tasks, as shown by evidence in cortical circuits [124], the impact on network computation of the plasticity of these synapses has been poorly investigated. 

In order to analyze the impact of IP onto neuronal network computation, researchers tend to employ circuits models with a limited number of parameters to control. The simplest network is composed of basic computational units, for instance, integrate and fire neurons, endowed with excitatory and inhibitory synapses randomly connected with a sparse architecture [125]. In such a model, the balance between excitation and inhibition controls a wide range of circuit dynamics, including average firing rate, and the way neurons respond collectively to inputs [125]. By assuming these circuits constraints, Vogels and colleagues in 2011 demonstrated that the implementation of an STDP rule at inhibitory synapses with strong potentiation in case of coincident pre- and postsynaptic spikes, allows inhibition to approach output firing rates to a target value [126]. The final target value strongly depends on the ratio between LTP and LTD, while deviations from setpoint are suppressed by a contrary reaction. The resulting effect is, therefore, the stabilization of firing rates whenever the incoming of repeated and persistent excitatory inputs tend to disrupt the E/I balance [127]. If network connectivity is organized with clusters of excitatory units, the E/I ratio is extremely sensitive to the wiring architecture. This, in turn, brings the network to a winner-takes-all behavior. The implementation of inhibitory plasticity in such a context, by compensating changes in firing rates, prevents groups of neurons from dominating the network [128]. Moreover, it has also been demonstrated that, in a network where connections are implemented with realistic connectivity patterns, and single neuron firing rates are sparse throughout the network, synaptic weights can be dynamically adjusted by inhibitory plasticity to equalize E/I balance changes [129]. The increasing complexity encountered in more organized circuits models like multiple layered feedforward networks strongly limits the capacity of inhibitory plasticity to maintain the E/I ratio. Nevertheless, Haas and colleagues showed that in the entorhinal cortex the strengthening of inhibitory connections can block the propagation of excitatory waves, ensuring network stability [107]. Inhibitory plasticity has also been suggested to favor the selection of specific feedforward pathways either by altering or by maintaining the balance between excitatory inputs and inhibitory signals [130]. In 2019 Wilmes and Clopath published a work where, by using a spiking model of layer 2/3 primary visual cortex, they showed that IPs play a major role in adjusting stimulus representation by storing information about reward stimuli. This model allowed to demonstrate that IP is essential to increase stimulus representation by triggering excitatory plasticity [131]. Moreover, in a recent study Soloduchin and Shamir showed that in a network model composed of two neuronal populations reciprocally inhibited, the implementation of a simple STDP rule for inhibitory synapses can bring the network to rhythmic activity by itself [132], confirming the hypothesis that, spontaneous oscillation can be entrained by modulation of peri-somatic inhibition on principal excitatory neurons [122]. Finally, it has been proposed that the spatial tuning patterns showing invariance and selectivity observed, for instance, in hippocampal place cells could be the result of excitatory and inhibitory plasticity. The combination, in fact, of the two mechanisms leads to localized activity invariant to different spatial dimensions [133]. 

Synaptic plasticity is also thought to be the cellular and molecular counterpart of learning and memory. In particular, memories that can be recalled by contextual cues or commands must involve plasticity mechanisms to be exploited [134]. In Hopfield networks, a circuit model assembled with recurrent connectivity and particularly suitable to implement associative memory, groups of silent and active neurons recruited by recalling inputs are used to describe memories [135]. Given the mixture of excitatory and inhibitory connections in such a network, it can be envisaged that IP has a leading role in creating and exploiting memories recall. In a different perspective, Maas and colleagues proposed that memories are represented by networks through the pathway generated by the population activity of the whole circuit, instead of activating a bunch of neurons [136]. In networks with strong and random excitatory recurrence, inhibitory plasticity stabilizes circuit dynamics similar to what happens in the motor cortex during the execution of limb movements. These networks, in fact, amplify the activity states that can be used to execute movement patterns [137]. By using a supervised learning scheme for feedforward and recurrent connections, Gilra and Gestner showed that IPs could efficiently accomplish linear, non-linear, or chaotic dynamics, as well as motor coordination dynamics [138]. Similarly, the implementation of STDP rules at different sites in a cerebellar like structure allowed to implement an efficient adaptive scheme capable of motor learning performance [139]. Finally, the specificity of inhibitory feedback sustaining grid cell organization has been suggested to require IP for the generation of grid cell population (Table 1) [140].

## 5. Perspectives and Concluding Remarks

The analysis of inhibitory plasticity presented so far is limited to the molecular characterization of the mechanisms underlying IPs together with the effects at the circuit level. However, a univocal determination of the roles for the IPs in circuit functions is still lacking, and evidences have been collected regarding the consequences of inhibitory plasticity at the integrative level. It is well known in fact that alterations of inhibitory circuits can contribute to the induction of neurological disorders. In particular, the regulation of the E/I balance, which is inherently bound to IP impacts the induction of excitatory LTP and LTD and can indeed shift the threshold required to switch between the two plastic conditions. It has recently been observed, in fact, that the inhibitory LTP, by modulating E/I balance, can effectively restore the hippocampal excitatory LTD preventing memory impairment related to Aβ protein accumulation [155]. Similarly, the disruption of the E/I balance in neocortical circuits is one of the most accredited explanation for the insurgence of the autism spectrum disorder (ASD), which can be therefore bound to inhibitory plasticity [156,157]. Again, schizophrenic patients often show disturbances in the GABAergic neurotransmission of the dorsolateral prefrontal cortex [158]. In particular, alterations in the peri-somatic regulation of pyramidal neurons lead to a reduced capacity of synchronizing gamma-band activity. Furthermore, one of the hypotheses on the etiology of epileptogenesis regards the hyperactivity of GABAergic neurons masking hyperexcitability activated by alterations of inhibitory strength triggered by insult of injury [159]. This cascade mechanism, responsible for the occurrence of the temporal lobe epilepsy, could be activated by overexpression or disruption of inhibitory plasticity. Finally, patients with Parkinson’s disease were shown to display downregulation of GABAergic activity in the afferents to basal ganglia. The reduction of neurological symptoms observed in response to deep brain stimulation, one of the most efficient therapeutic treatments, can be attributed to the triggering of GABAergic LTP allowing the recovery from the aforementioned GABA downregulation [160].

Besides the obvious interest in the biomedical and clinical fields, the analysis of brain functions and particularly of synaptic connections also arouse great interest in computer science and, more generally, in the field of neuromorphic electronic and artificial intelligence. The goal in fact to build large artificial neural networks with vast amounts of computing elements has rendered the task of creating low consuming artificial synapses a high priority. The visionary idea of electrical elements behaving like synapses called memristors is now becoming true [161]. These physical devices can effectively behave like synaptic elements because of their capacity to reproduce synaptic features and plastic mechanisms such as STDP [162] or heterosynaptic plasticity [163]. Memristors can indeed be assembled to mimic neural functions and reproduce neuronal behavior [164] or perform autonomous complex learning tasks [165]. Nowadays, several physical elements have been proposed and adopted to reproduce synaptic learning; however, none of them has been used to mimic plastic behaviors of inhibitory synapses. There is therefore increasing attention on the role of the various forms of IP in circuit computation and on the further possibility to introduce such behaviors into electronic circuits performing complex tasks. It can be envisaged in fact that the implementation of unsupervised learning rules in electronic synapses could entrain artificial circuits to perform autonomous behavior [165]. The growing expansion of the neuromorphic field and of the brain-inspired computation requires to implement devices exploiting the properties of both excitatory and inhibitory synapses. The encoding then of learning rules in artificial synapses and in electronic circuits, paving the way to the next generation of neuromorphic devices, is opening promising perspectives for a series of applications with clinical relevance such as neuroprosthesis or with a high social impact for daily lives driving futuristic artificial intelligence machines.

## Figures and Tables

**Figure 1 ijms-21-01805-f001:**
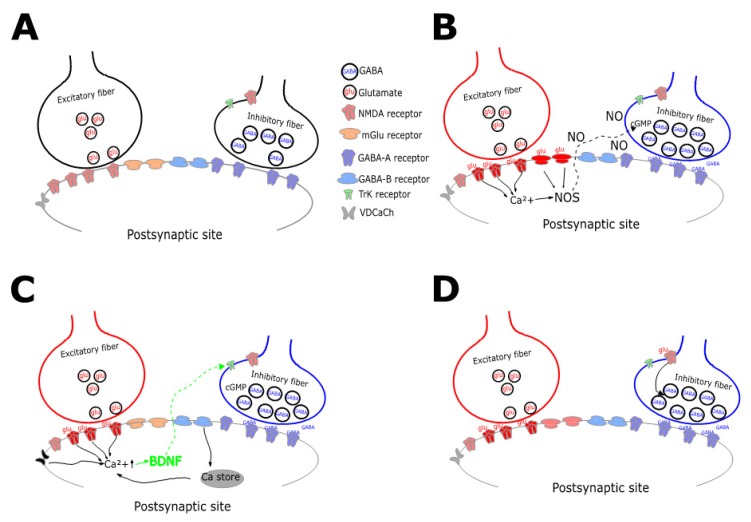
Schematic diagram collecting mechanisms underlying presynaptic LTP. A. Excitatory and inhibitory fibers contact, by releasing glutamate (glu) and GABA (GABA), a postsynaptic neuron expressing both ionotropic (mglu) and metabotropic receptors (GABA-B). B The repetitive release of glutamate triggers calcium entry through NMDA receptors in the postsynaptic terminals. The activation of Nitric Oxide (NO) synthase (NOS) induces the retrograde diffusion of NO which, in turn, activates cyclic-GMP potentiating vesicles release. C. Similarly to B, the repetitive glutamate release causes postsynaptic intracellular calcium rise in response to i) Voltage-dependent Calcium Channels (VDCaCh) opening, ii) NMDA receptors opening, or iii) mGlu receptors activation causing release from intracellular stores. Calcium increase triggers the retrograde diffusion of the Brain-Derived Neurotrophic Factor (BDNF) potentiating GABA release via Tyrosine Kinase-1 (TRK) receptors activation. D. The diffusion of glutamate in the extrasynaptic space can directly activate presynaptic NMDA-Rs favoring the potentiation of GABA release.

**Figure 2 ijms-21-01805-f002:**
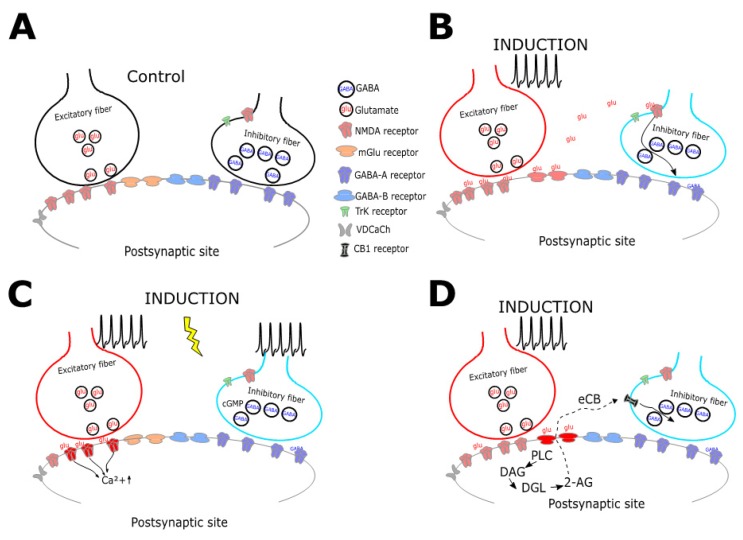
Schematic diagram collecting mechanisms underlying presynaptic LTD. A. Excitatory and inhibitory fibers contact, by releasing glutamate (glu) and GABA (GABA), a postsynaptic neuron expressing both ionotropic (mglu) and metabotropic receptors (GABA-B). B The diffusion of glutamate in the extrasynaptic space can directly activate presynaptic NMDA-Rs inducing the depression of GABA release. C. The coactivation of glutamatergic and GABAergic ionotropic receptors by simultaneous stimulation of excitatory and inhibitory fibers can lead to the depression of GABA release via a not well-identified mechanism. D. The activation of metabotropic glutamate receptors following repetitive excitatory stimulation triggers intracellular signal cascade, typically involving Phospholipase-C (PLC), diacylglycerol (DAG), Diacylglycerol lipase (DGL) and the 2-Arachidonoylglycerol (2-AG) endocannabinoid (eCB). This class of molecules can freely diffuse in the extracellular space acting as a retrograde messenger to activate specific cannabinoid receptors (CB) onto the GABAergic terminal that trigger the depression of vesicles release via different pathways.

**Figure 3 ijms-21-01805-f003:**
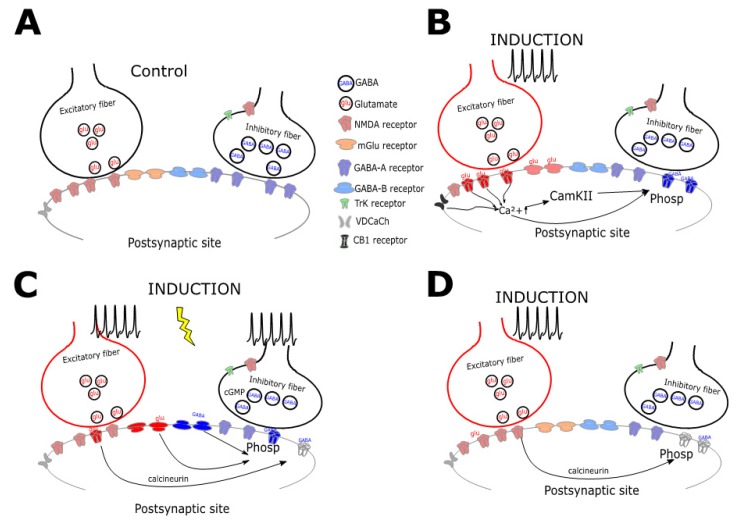
Schematic diagram collecting the mechanisms underlying postsynaptic plasticity. A. Excitatory and inhibitory fibers contact, by releasing glutamate (glu) and GABA (GABA), a postsynaptic neuron expressing both ionotropic (mglu) and metabotropic receptors (GABA-B). B. The activation of glutamatergic synapses can lead to an increase of postsynaptic intracellular calcium concentration either through NMDA-Rs or VDCaChs opening. Calcium increase can directly act on proteins phosphorylation (phosp) or can mediate CamKII activation leading to phosphorylation as well. Postsynaptic GABA-A receptors can thus increase their efficacy, can switch from silent to active state, or can move in the postsynaptic membrane. C. The simultaneous activation of metabotropic glutamatergic and metabotropic GABAergic receptors can lead to the phosphorylation required for the potentiation of ionotropic receptors activity. D. Conversely, the reduction of GABA-A-Rs receptors activity, either in the form of receptors silencing or in the sliding away from the postsynaptic density, can be induced by NMDA receptors opening following glutamate released. The protein phosphatase calcineurin or alternatively, the increase of intracellular calcium concentration in the postsynaptic neurons through NMDA or VDCaChs are the mediators of postsynaptic i-LTD.

**Table 1 ijms-21-01805-t001:** Mechanisms of inhibitory plasticity and functional consequences.

Sign of Plasticity	Molecular Mechanism	Brain Region/Neuron	Site of Expression	Computation/Functional Significance	Refs
LTP	GABA_B_ receptor dependent, BDNF signaling	Visual cortex/Neonatal hippocampus	Presynaptic	Critical period plasticity/ E/I balancing	[54,104,105,141]
LTP	Postsynaptic NMDA, retrograde NO	VTA, Basolateral amygdala Cerebellum,	Presynaptic	Reward modulation/ spatio temporal pattern sharpening/ shaping conditioned fear response	[44,46,142]
LTP	Postsynaptic calcium, retrograde BDNF	Hippocampus	Presynaptic	Associative memory formation	[143]
LTP	Presynaptic NMDA	Cerebellum	Presynaptic	Motor learning regulation	[65,144]
LTP	Postsynaptic mGluR and retrograde NO	Lamina I spinal cord	Presynaptic	Signal to noise regulation	[53]
LTP	Postsynaptic calcium/NMDA	Deep cerebellar nuclei	Presynaptic	Regulation of spike firing for motor coordination	[145,146]
LTP	Postsynaptic NMDA and CamKII	Medial prefrontal cortex	Postsynaptic	Local regulation of E/I at cellular level	[147]
LTP	Postsynaptic Calcium/ CamKII	Cerebellum Purkinje cell	Postsynaptic	Regulation of output firing patterns	[52,141,148]
LTP	GABA_B_/ mGluR	Hippocampal CA1	Postsynaptic	Reinforcement of rhythmic activity	[149]
LTP	Presynaptic firing paired with mild depolarization	Developing visual cortex	Postsynaptic	Regulating critical period for ocular dominance	[76]
LTP	Calcium influx receptor phosphorilation	Neocortex	postsynaptic	E/I balancing	[83,84,85]
LTP	Postsynaptic NMDA and calcium rise	Lateral amygdala	Postsynaptic	Processing stimuli during fear conditioning	[150]
LTP	Postsynaptic NMDA, L type calcium channels	Auditory cortex	Postsynaptic	Normalizing E/I and remodeling auditory map	[108]
LTD	mGlur, retrograde eCB	Hippocampus, amygdala, Visual cortex, prefrontal cortex	Presynaptic	Changes of E/I / extinction of aversive memories/ regulation of development in critical period	[49,69,72,75]
LTD	GABA_A_ activation and postsynaptic NMDA	Neonatal hippocampus	Presynaptic	Regulation of synapse formation and maturation	[151]
LTD	Presynaptic NMDA	Cerebellum, visual cortex	Presynaptic	Spatio-temporal sharpening sensory information	[46,77]
LTD	Postsynaptic NMDA and mediated by calcineurin	hippocampus	postsynaptic	Disinhibit excitatory circuits	[152]
LTD	Postsynaptic calcium and protein phosphatase	Deep cerebellar nuclei	postsynaptic	Modulation of spontaneous cerebellar firing for motor coordination	[153]
LTD	Dopamine mediated eCBN signaling	Ventral tegmental area	postsynaptic	Regulation of addiction mechanisms	[75,154]

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
