# Peer review of "Inhibitory Plasticity: From Molecules to Computation and Beyond"

_ijms, 2020, doi:10.3390/ijms21051805_

Round 1

Reviewer 1 Report

This is a very well written manuscript by Gandolfi et al., which deeply analyze the mechanisms of inhibitory plasticity, processes of increasing relevance in neurological diseases.

This referee found very interesting the analysis of the data as well as the careful and critical discussion carried out by the authors. The authors have shown they have a wide knowledge of the field, which is reflected in the structure of the paper. My enthusiasm was growing reading the paper. However, I have 3 advices to improve the paper:

It worth to add a more detailed discussion related to long term potentiation of synaptic inhibition mediated by metabotropic mechanisms. You have included work describing LTP of slow synaptic inhibition in vitro (Huang CS et al, Cell. 2005 Oct 7;123(1):105-18) but there are two additional papers in PNAS in 2009 from the same group authored by Chung HJ that should be discussed. They highlighted the relevance of GirK channels in LTP of synaptic inhibition. Additionally, very recently it has been demonstrated for the first time the importance of such mechanism in vivo (Sanchez-Rodriguez et al., Int J Mol Sci. 2019 Mar 7;20(5)), as the LTP of synaptic inhibition mediated by GirK channels appears 48 hours after high frequency stimulation in order to assure the reset of the system to generate a new LTP.

The control of excitation/inhibition balance has been shown to be related to the threshold for LTP/LTP induction, accordingly to BCM Theory. It would be interesting to include some discussion about the theory, specially when these mechanisms has been related to neurological disorders, as in Alzheimer models it has been shown HFS-induced LTD instead LTP by modulation of the synaptic plasticity induction threshold (Moreno-Castilla et al., 2016, doi: 10.1016/j.neurobiolaging.2016.02.021; Sanchez-Rodriguez et al., 2019 doi: 10.1111/jnc.14946).

Finally, both figures have very poor quality: Letter sizes, color, adjustments, etc…In the same way, figure legends include very scarce information. Figures are not in accordance with the quality of the main text. Please, improve them.

Author Response

We thank the reviewers and editor for careful analysis of the manuscript. Based on reviewers’ comments, changes to the manuscripts have been introduced.

The manuscript has been improved following the comments raised by reviewers, furthermore a few changes throughout the text have been done. In particular, at line 189 page 8 a new sentence has been added. Moreover, three new references were included in the text

Below reviewers’ concerns are addressed point by point

We thank the reviewer for his/her helpful comments

  • It worth to add a more detailed discussion related to long term potentiation of synaptic inhibition mediated by metabotropic mechanisms. You have included work describing LTP of slow synaptic inhibition in vitro (Huang CS et al, Cell. 2005 Oct 7;123(1):105-18) but there are two additional papers in PNAS in 2009 from the same group authored by Chung HJ that should be discussed. They highlighted the relevance of GirK channels in LTP of synaptic inhibition. Additionally, very recently it has been demonstrated for the first time the importance of such mechanism in vivo (Sanchez-Rodriguez et al., Int J Mol Sci. 2019 Mar 7;20(5)), as the LTP of synaptic inhibition mediated by GirK channels appears 48 hours after high frequency stimulation in order to assure the reset of the system to generate a new LTP.

The reviewer properly raised the issue of GirK channels contribution in the regulation of inhibitory plasticity. A paragraph taking into account for that issue has been inserted at line 279 page 12.

  • The control of excitation/inhibition balance has been shown to be related to the threshold for LTP/LTP induction, accordingly to BCM Theory. It would be interesting to include some discussion about the theory, specially when these mechanisms has been related to neurological disorders, as in Alzheimer models it has been shown HFS-induced LTD instead LTP by modulation of the synaptic plasticity induction threshold (Moreno-Castilla et al., 2016, doi: 10.1016/j.neurobiolaging.2016.02.021; Sanchez-Rodriguez et al., 2019 doi: 10.1111/jnc.14946).

Again, the reviewer raised an interesting point regarding the functional and physiopathological effects of inhibitory plasticity. In particular, careful attention should be paid to the changes of the threshold required to switch between LTP and LTD observed in neurological diseases.

The perspective and concluding remarks section has been rephrased and a specific paragraph has been considered carefully.

  • Finally, both figures have very poor quality: Letter sizes, color, adjustments, etc…In the same way, figure legends include very scarce information. Figures are not in accordance with the quality of the main text. Please, improve them.

We apologize for the poor figure quality. Based on reviewer’s concern, we have changed the overall figure layout which is now organized with three pictures instead of two diagrams summarizing the principal mechanisms underlying inhibitory plasticity described in the manuscript.

Reviewer 2 Report

In this manuscript, the authors reviewed and discussed the most recent findings of diverse forms of inhibitory plasticity in different brain areas. They discuss how the various forms of inhibitory plasticity shape brain circuits architectures and functions. They also reviewed various molecular pathways involved in the induction and expression rules of inhibitory plasticity.  This is very interesting review and it is well structured and written with a good standard of English language. The work is worthy of consideration for publication, there is just one typo error in Page 5, line 123, It appears thus evident that, remove thus.

Author Response

We thank the reviewers and editor for careful analysis of the manuscript. Based on reviewers’ comments, changes to the manuscripts have been introduced.

The manuscript has been improved following the comments raised by reviewers, furthermore a few changes throughout the text have been done. In particular, at line 189 page 8 a new sentence has been added. Moreover, three new references were included in the text

Below reviewers’ concerns are addressed point by point

1 – Page 5 Line 123: the typo error.

Corrected and highlighted in red in the revised version of the manuscript

Round 2

Reviewer 1 Report

The paper has been improved. The figures are now appropiated but please, enlarged them to 2 columns wide.